# Neuroendocrine Modulation of the Immune Response after Trauma and Sepsis: Does It Influence Outcome?

**DOI:** 10.3390/jcm9072287

**Published:** 2020-07-18

**Authors:** Philipp Kobbe, Felix M. Bläsius, Philipp Lichte, Reiner Oberbeck, Frank Hildebrand

**Affiliations:** 1Deparment of Trauma and Reconstructive Surgery, University Hospital RWTH Aachen, D-52074 Aachen, Germany; p.kobbe@ukaachen.de (P.K.); fblaesius@ukaachen.de (F.M.B.); plichte@ukaachen.de (P.L.); 2Deparment of Trauma and Hand Surgery, Wald-Klinikum, 07548 Gera, Germany; reiner.oberbeck@wkg.srh.de

**Keywords:** trauma, sepsis, DHEA, steroids, catecholamine

## Abstract

Although the treatment of multiple-injured patients has been improved during the last decades, sepsis and multiple organ failure (MOF) still remain the major cause of death. Following trauma, profound alterations of a large number of physiological systems can be observed that may potentially contribute to the development of sepsis and MOF. This includes alterations of the neuroendocrine and the immune system. A large number of studies focused on posttraumatic changes of the immune system, but the cause of posttraumatic immune disturbance remains to be established. However, an increasing number of data indicate that the bidirectional interaction between the neuroendocrine and the immune system may be an important mechanism involved in the development of sepsis and MOF. The aim of this article is to highlight the current knowledge of the neuroendocrine modulation of the immune system during trauma and sepsis.

## 1. Introduction

Despite profound improvements in the initial care and in the treatment of multiple injured patients that follows, MOF and sepsis represent an ongoing threat [1,2]. It is assumed that the posttraumatic changes of the immune system crucially contribute to the development of these complications in multiple-injured patients. This includes pro- and anti-inflammatory changes of the immune system while an excessive reaction of both of the components leads to a massive disturbance of the immunological homeostasis [2,3,4].

Parallel to the changes of the immune system in multiple-injured as well as in septic patients, neuroendocrine systems are activated. Activation of the sympatho-adrenergic system (SAS) leads to a massively increased release of the catecholamines adrenaline and noradrenaline into the circulation [5,6]. The released adrenaline mainly originates from the adrenal medulla, and the noradrenaline mainly originates from the postganglionic sympathetic nerve fibers [6]. An increased release of catecholamines occurs in the initial stage as well as in the acute and late stage of sepsis and is enhanced by the released pro-inflammatory cytokines (Interleukin (IL)-6, Tumor Necrosis Factor (TNF)-α) [6]. Furthermore, the spleen, the lung and the gut-associated lymphoid tissue (GALT) are tightly sympathetically innervated and play a crucial role with respect to the adrenergic modulation of the immune system. In addition, it could be demonstrated that most of the cells of the immune system are equipped with α- as well as β-adrenergic receptors on their cell surface and that many of these cells are able to synthesize catecholamines themselves [7].

Apart from the activation of the SAS, a massive release of hormones of the hypothalamic–pituitary–adrenal axis (HPA-axis) or the hypothalamic–gonadal axis (HPG-axis) is found [5,6]. Activation of the HPA-axis is detectable after severe traumata as well as in septic patients and is responsible for a massive increase of cortisol and its release hormone ACTH. Here, the stimulation of the HPA-axis by pro-inflammatory cytokines like TNF-α, IL-1 and IL-6 plays a crucial role [8]. Between the amount of the cortisol level and the severity of the illness, a positive correlation exists. In cases of a prolonged course of disease, a peripheral glucocorticoid resistance develops characterized by normal or decreased ACTH and elevated cortisol levels [6].

With regard to the HPG-axis, which is likewise controlled by the release of hormones of the hypothalamus, decreased testosterone levels could be found in men after severe trauma and during sepsis whereas women react with an increase of their estrogen levels, presumably based on an increased aromatizing of androgens. In this case, it also comes to an influence of pro-inflammatory cytokines on hormone release [9].

Blood levels of the steroid hormone Dehydroepiandrosterone (DHEA) and its sulphated pattern (DHEA-S) are significantly decreased in critically ill and septic patients [10,11]. DHEA is the quantitatively most important human steroid hormone, which is produced mostly in the adrenal gland but also in the gonads. DHEA has not only a potent immunomodulatory activity by itself, but it is also considered to be a precursor of the androgen and estrogen biosynthesis [12]. During sepsis and trauma, a dissociation of cortisol and DHEA is found, which leads to an imbalance between immune-suppressive and immune-stimulating steroid hormones [8,13]. In accordance with this, it was shown that depressed levels of circulating DHEA in patients with sepsis are positively correlated to the risk of death [14,15].

Until now, it was assumed that the activation of neuroendocrine systems (SAS, HPA-axis, HPG-axis, DHEA) during trauma and sepsis serves the adaption of physiological systems like metabolism, heart/circulation, tissue regeneration and the central nervous system onto the elevated requirements.

The aim of this review is, on the one hand, to highlight current insights on how neuroendocrine released messengers are responsible for immunomodulation following severe trauma and during sepsis and, on the other hand, to find out whether this knowledge has been transferred into clinical practice.

## 2. Hypothalamic–Pituitary–Adrenal (HPA) Axis

Trauma and sepsis cause complex alterations of the hypothalamic–pituitary–adrenal axis and glucocorticoid signaling [16]. The immunomodulatory effects of glucocorticoids are well described. On the one hand, glucocorticoids inhibit the release of pro-inflammatory cytokines from T helper-1 (Th1) and antigen-presenting cells (APCs), and on the other hand, glucocorticoids induce the release of anti-inflammatory cytokines from T helper-2 (Th2) cells [17]. Through thismechanism glucocorticoids cause a Th1/Th2-shift of the immunological response. Additionally, glucocorticoids inhibit the function of neutrophil and eosinophil granulocytes as well as macrophages [18,19]. Glucocorticoid-induced immune suppression is used in autoimmune disorders and after organ transplant, and theoretically, these immunomodulatory effects may attenuate the overwhelming inflammatory response following severe trauma or during sepsis. Especially under consideration that critical ill patients show signs of relative adrenal insufficiency by suppression of the HPG axis due to the critical illness as well as sedative/analgesic drugs [16,20], the systemic substitution of glucocorticoids and the modulation of the metabolism of glucocorticoids appears reasonable. Recently, the role of 11β-Hydroxysteroid dehydrogenase type 1 (11β-HSD1) in acute and chronic inflammation has been pointed out [21,22,23]. 11β-HSD1 causes an intracellular conversion of inactive cortisone to the active cortisol. Therefore, 11β-HSD1 is an intracellular gate-keeper for glucocorticoid action [22]. Interestingly, the expression of 11β-HSD1 is greatly up-regulated during differentiation of monocytes into macrophages thus theoretically curbing the inflammatory potency of these cells [21]. However, it appears that this intracellular immunomodulation by 11β-HSD1 is disturbed during trauma and hemorrhage resulting in an inefficacy of released glucocorticoids to modulate the inflammatory response [24].

There are numerous animal studies in which corticosteroid administration consistently protected against lethal sepsis; however, clinical trials in sepsis found much less consistency in survival benefits from corticosteroids, although most trials demonstrated faster resolution in shock and organ dysfunction [25]. The Corticosteroid Therapy of Septic Shock (CORTICUS) study showed no benefit of hydrocortisone on survival or reversal of shock in patients with septic shock [26]. Although hydrocortisone treatment of patients in septic shock resulted in faster improvement of organ function, mainly of the cardiovascular system, this had no effect on mortality [27]. The HYPOLYTE (Hydrocortisone Polytraumatise) study showed that in intubated trauma patients hydrocortisone significantly reduced the risk of hospital-acquired pneumonia, however, again without altering the mortality rate [28]. Critics of glucocorticoid application argue that acute-phase sepsis is associated with increased glucocorticoid receptor expression and cortisol concentrations, possibly implying no need for exogenous substitution, which may even increase glucocorticoid resistance through a negative feedback mechanism [29]. Furthermore, glucocorticoid application is known to have potential hazardous side-effects, especially a slight increase of the incidence of clinically important gastrointestinal bleeding in critical ill patients [30].

Despite above mentioned findings of single studies, recent data from meta-analyses suggests with a low- to moderate-quality evidence that a long course of low-dose corticosteroids reduces 28-day mortality without inducing major complications [25,31,32,33]. Nonetheless, corticosteroid therapy for septic and trauma patients remains controversial despite general agreement that corticosteroids improve sepsis-associated comorbidities, such as shock, organ dysfunction, and length of hospital stay.

An immunomodulatory effect is also found for other hormones of the anterior pituitary gland as ACTH, β-endorphin, and prolactin. The administration of prolactin during experimental polymicrobial sepsis in mice increased the mortality rate from 47% to 81%. This effect was paralleled by a significant increase of splenocyte apoptosis rate and a marked depression of splenocyte proliferation. Furthermore, prolactin administration profoundly affected cellular cytokine release (IL-2, IL-6, IFN-γ) 48 h after induction of sepsis by cecal ligation and puncture [34]. However, the available literature about the immunomodulatory effects of pituitary hormones is controversial despite the suppressive effects on the immune cell proliferation and activity or rather the release of pro-inflammatory cytokines predominantly shown (Table 1) [35].

## 3. Hypothalamic–Pituitary–Gonadal (HPG) Axis

A relevant interaction between the HPG-axis and severe trauma and infections has been well described. On the one hand, traumatic and infectious insults induce a central suppression of the HPG axis with reduced gonadal androgen production and an associated state of catabolism (reduction of muscle mass, increased nitrogen loss) [36]. Therefore, the therapeutic use of a synthetic androgen (oxandrolone) has been investigated to support the switch to an anabolic situation. However, in contrast to burn injuries, it was not found to exert beneficial effects after major trauma [37]. On the other hand, sex hormones have the potential to significantly influence the posttraumatic course [11]. The so-called gender-dependent dimorphism of morbidity and mortality after trauma and sepsis is mainly based on experimental data indicating that sex hormones have relevant effects on various organs and the immune system after trauma and sepsis [38]. In general, 17-β-estradiol (E2) and estrogen-receptor agonists were shown to have beneficial effects on organ function, tissue damage (e.g., neutrophil infiltration, edema formation) and the immune response, whereas androgens (e.g., testosterone) were associated with enhanced organ damage and a dysfunctional immune system [39,40,41,42]. For example, E2 was shown to be beneficial for cardiac function after trauma-hemorrhage (TH) (e.g., increased cardiac output) due to protective effects on myocardial mitochondria, induction of heme oxygenase-1 (HO-1) [43], downregulation of nuclear factor kappa-light-chain-enhancer of activated B-cells (NF-κB) and the modulation of inflammatory mediators, such as heat shock proteins (HSP) and cytokines. In this context, E2 decreased the likelihood of cardiomyocyte necrosis by increasing HSP70 levels [44]. Furthermore, an E2-associated reduction of pulmonary damage after TH with a decrease of edema and neutrophil infiltration was observed [45,46]. As potential mechanisms for the beneficial hepatic effects of E2, a restoration of liver metabolism (normalization of adenosine triphosphate) and a reduction of apoptosis by up-regulation of Bcl-2 and induction of HO-1, heat shock proteins (HSP) and p38 mitogen activated protein kinase (p38 MAPK) was discussed. E2 attenuated acute renal injury by reducing apoptosis, endothelial cell damage and inflammation. Moreover, the intestine benefits from posttraumatic E2-application due to an increased blood flow and a decrease of neutrophil infiltration [45,47,48]. Furthermore, sex hormones significantly modulated cellular and humoral immune functions after TH. Interestingly, enzymes relevant for the synthesis of sex steroids were also found in immune cells (e.g., T-cells). The expression and activity of these enzymes were modulated by TH and differed between males and proestrus females. In males, TH resulted in increased levels of 5α-reductase which catalyzes the synthesis of 5α-dihydrotestosterone (5α-DHT). 5α-DHT itself is well known to exert even more pronounced immunosuppressive effects than testosterone. In contrast, in proestrus females, an enhanced aromatase activity was observed which resulted in an increased E2 synthesis [49]. Thus, these changes might contribute to the gender-related differences found after TH and sepsis [40]. Focusing on specific immune cells, macrophages from different tissue compartments were shown to respond differently following TH with an increased productive capacity for TNF-α and IL-6 of hepatic (Kupffer cells) and alveolar macrophages and a decreased synthesis of splenic and peritoneal macrophages [40]. Administration of E2 resulted in a normalization of the productive capacity but also in a beneficial modulation of toll-like receptor (TLR) and iNOS synthesis [48]. Furthermore, both dendritic (reduced antigen presentation) and T-cell function (shift from T-helper (Th) 1 to Th2-cell) were impaired after TH, which was assumed to contribute to the increased susceptibility for infectious complications after trauma. Again, these changes were especially pronounced in males and were reversed by E2-treatment [50,51]. In addition, an E2-induced interference with the apoptosis rate of immune cells and the synthesis of HSP was verified [2,39,51].

It has been proven that these effects of sex hormones are mediated by specific androgen- (AR) and estrogen (ER) receptors, which are expressed by almost all cells. For AR no further major subdivision is reported, whereas ER are subdivided in two major subtypes, ERα- and ER-β. The distribution of these ER is organ specific. In both, lung and heart, the effects of E2 seem to be mediated via ER-β [52]. In the liver, ER-α dependent effects were primarily described; however, ER-α-independent effects were also found [53]. Moreover, in the intestine, ERα- and ER-β mediated effects of E2 were found [54]. Hepatic and splenic macrophages and T-cells seem to preferably express ER-α and show improved function after administration of E2 or the selective ER-α-agonist propyl pyrazole triol (PPT) [42], whereas alveolar macrophages normalized posttraumatic function after application of a ER-β-agonist [40]. Based on the aforementioned results, it seems likely that ER subtypes may have tissue-specific roles in mediating the effects of E2 after TH.

All these aforementioned, beneficial effects on the various organs and the immune system might contribute to the improved survival observed after E2-application in animal models of TH even without large volume fluid resuscitation. In these studies, an E2-associated shift of the remaining blood into heart, liver and kidney has been discussed as a further favorable aspect [55,56,57].

Although sex hormones have been described to also influence human immune cell function, knowledge about these effects after TH and sepsis is relatively sparse. After infections, human mononuclear cells from males were shown to produce lower levels of type I interferons (IFN) in response to TLR-7 ligands and higher IL-10 in response to TLR-9 ligands as compared to females [58]. Furthermore, women were shown to have an enhanced capability of producing antibodies [59,60]. Clinical studies also described effects on the humoral immune response after multiple trauma and sepsis [61,62]. In this context, females younger than 50 years with an ISS > 25 had significantly lower plasma cytokines after multiple trauma [62].

Furthermore, diverse clinical studies found evidence for a more favorable clinical course after trauma and sepsis for females. In this context, female gender was associated with a lower incidence of posttraumatic infections (e.g., pneumonia, sepsis) and multiple organ failure (MOF) [63,64,65,66,67,68]. As a result, less requirements for intensive care were associated in female patients [69].

In some studies, however, only premenopausal females (in most studies defined as <50 years) showed lower incidences of posttraumatic complications (e.g., sepsis, MOF, mortality), a reduction of lactate levels and decreased blood transfusion requirements compared to males of the same age [62,65,66,70,71,72,73]. This might support the findings of the aforementioned experimental studies indicating a beneficial effect of E2 on the further clinical course. However, both, reduced sepsis rates and survival benefits have also been described for postmenopausal women [61,74,75]. Here, persistently increased androgen levels and the associated immunosuppression in males might play a role.

In the search for the mechanisms of the immunomodulating effect of estrogen, myeloid-derived suppressor cells (MDSC), a heterogeneous population of the myeloid lineage, which modulate the adaptive immune response, have attracted the focus of research efforts [76]. These cell subtypes have been shown to constitute a crucial component of the innate immune system in various inflammatory states. During systemic inflammation, MDSCs are recruited from the bone marrow [77]. In this respect, it could be shown in different mouse models that after CLP procedure and recruitment, an accumulation of MDSCs takes place in secondary lymphatic organs. The recruitment of MDSCs is mediated by MAMPs (e.g., LPS) and DAMPs (e.g., HMGB1) [77,78,79]. Furthermore, MDSC are induced and activated in the presence of estrogen and cytokines, such as IL-6, IFN-γ and IL-1β, and strongly contribute to T-cell dysfunction in various diseases such as sepsis, tumorigenesis and trauma [80,81,82]. Especially in the case of estrogen, a direct activation of the STAT3 signaling pathway and upregulation of JAK2 and SRC in MDSCs and a consecutive anti-inflammatory function of MDSCs could be demonstrated [83].

However, results on sex-related differences after trauma and sepsis are not unequivocal. Some studies found no differences for complications (e.g., sepsis) and mortality after blunt trauma [62,67,70,84,85], whereas even increased mortality rates for females with an age around 80 years were observed by Eachempati et al. [86]. Without specifically considering patient’s age, also others described an increased mortality rate in females after development of infections (e.g., pneumonia, sepsis) or after major surgery [86,87]. In a study of Dossett et al. increasing levels of E2 were even associated to a higher mortality rate of critically ill patients [88].

There might be different reasons for these partly divergent results. Despite significant effects of the estrous cycle on E2 levels, none of the aforementioned studies has determined the cycle phase, some did not consider age. In addition, the effects of co-morbidities, pre-medication (e.g., oral contraceptives, hormone replacement therapy), injury severity and distribution (e.g., traumatic brain injury, TBI) have not been considered so far. In this context, it is well known that TBI has the potential to depress systemic E2 levels; however, this association has not been taken into account in most of the studies [89]. Consideration of patients with mild or moderate trauma and a low risk of adverse outcome might keep gender-related effects from becoming visible [65]. In this regard, diverse studies found gender-associated divergences only in patients with a high overall injury severity [62,73,90].

In conclusion, particularly based on experimental but also indirectly from clinical studies [62,65], it is likely that not the gender itself, but sex hormones influence the immune response, the incidence of complications and the outcome after trauma and sepsis [9,89]. Therefore, consideration of the sex hormone status could be an important step for an individual therapeutic approach after trauma. Therapeutic utilization of the interaction of E2/testosterone with the immune system after trauma and sepsis may offer new strategies. These include the application of E2, ER-agonists, androgen receptor antagonists and α-reductase-inhibitors, which prevent the conversion of testosterone into highly active dihydrotestosterone. However, up until now, no clinical trials have been published investigating the clinical value of these approaches (Table 1).

## 4. Dehydroepiandrosterone (DHEA)

DHEA, the most abundant circulating steroid in humans is mainly synthesized in the adrenal cortex. It has been well described to be involved in the response to trauma and sepsis [91]. Beside trauma- and sepsis-related effects on DHEA-synthesis and -metabolism, an immunomodulatory potency of DHEA under the aforementioned conditions has been proven in experimental studies. In this context, a beneficial effect of DHEA on the incidence of complications, the clinical status, survival and immune function after hemorrhagic shock, infections and a combination of these insults has been described [13,92,93]. Furthermore, an improved function of the cellular immune system, which includes a marked improvement of the proliferation rate of lymphocytes with a simultaneous decrease of the apoptosis rate of lymphocytes, was observed. At the same time, a varied pattern of cytokine release with a DHEA-induced inhibition of TNF-α, IL-6 and IL-10 release was found [13,92]. Accordingly, administration of DHEA immediately before induction of TH normalized immune cell migration (NK-cells, CD4+ and CD8+ lymphocytes) and splenocyte apoptosis rate in a murine model of hemorrhagic shock [94]. In a murine model of bilateral femoral fractures DHEA suppressed the serum levels of proinflammatory cytokines (IL-6, TNF-α, MCP-1) and also of IL-10 but did not improve markers of pulmonary inflammation [95]. Same effects of DHEA were found in case of neuroinflammation after TBI. In addition, DHEA resulted in improved long-term cognitive and behavioral outcome [96]. DHEA-treatment after infections displayed an increased survival rate, reduced bacterial contamination in the peritoneal fluid, decreased pro-inflammatory (e.g., TNF-α, IL-6) and enhanced anti-inflammatory cytokine release (e.g., IL-4, IL-10). These DHEA-associated changes of the inflammatory response were supposed to be caused by suppressed NO-secretion and a shift towards the Th2 response. Furthermore, an improved delayed-type of hypersensitivity reaction after DHEA-application was observed, demonstrating that apart from the innate immune system, also a modulation of the acquired part of the immune system is initiated [97,98]. In contrast, administration of DHEA in a murine sepsis model of cecal ligation and puncture revealed that there was no difference in the survival rate, the cellular proliferation and apoptosis rate whether mice received DHEA immediately before induction of a polymicrobial sepsis or after the development of the first septic symptoms [97]. The way of DHEA administration might represent a potential reason for these different findings. In this context, it was shown that subcutaneous administration of DHEA was accompanied by an improved survival and normalized immune functions whereas intravenous or intraperitoneal treatment with DHEA failed to exert the expected effects on the survival rate and immune functions of septic mice. Moreover, a varyingly strong activation of the HSP-70 release in the different shock organs (lunge, liver, kidney) could be detected in septic mice dependent on the way of DHEA administration [99].

However, it is of major relevance to note that transfer of animal data to the human situation has to be interpreted with caution. In this context, rodent adrenal DHEA production is modest compared to that of humans. Furthermore, these animals have the ability to convert exogenous DHEA to sex steroids. Therefore, studies specifically focusing on the human situation are of upmost importance. However, in vitro and clinical studies on the effects of DHEA on immune cells after trauma and sepsis are rare. In an in vitro study, DHEA-S was able to stimulate the synthesis of reactive oxygen species (ROS) via NADPH oxidase activation directly and thereby improve neutrophil function [100]. Neutrophils hold a unique position among the leukocytes, since they are the only subpopulation with an active transporter, the organic anion-transporting polypeptide D (OATP-D). Furthermore, neutrophils do not have steroid sulfatase, which activates DHEAS to DHEA. The effect of DHEA consequently must be a direct one. In addition, in case of primary adrenal insufficiency with a frequently associated deficit of DHEA/DHEAS, an impaired natural killer cell cytotoxic function was found. However, this impaired function was not influenced by longstanding DHEA-therapy [101]. Apart from this direct influence of DHEA on immune cells, an intracellular metabolization into sex steroids was postulated. In this respect, DHEA related activation of monocytes and their interaction with endothelial cells was shown to depend on the conversion to androgens and subsequent binding on androgen receptors [102]. DHEA may also antagonize the effects of glucocorticoids but also act as an inhibitor of the glucose-6-phosphatase in the hyperglycemic environments that are common after trauma [103]. Additionally, it was shown that cells of the immune system are able to synthesize DHEA, which could influence the cell function in an autocrine control loop [13].

In this context, it could be shown that the GR receptor is also expressed on circulating MDSCs. Lu et al. could show in an experimental study on a liver injury mouse model that the modulating effect of glucocorticoids is caused by suppression/activation of HIF1α and HIF1α-dependent glycolysis [104]. Based on these observations, the modulation of MDSC function by systemic steroids may represent a new therapeutic target, although detailed data on the timing of use and type of steroid are still lacking. Furthermore, the function of MDSC in sepsis and trauma has not been sufficiently studied [105,106]. The vast majority of data are based on studies in tumors, although an increasing number of studies highlight the role of MDSC subtypes in the resolution of inflammation after severe sepsis and trauma [106,107].

In the clinical setting, DHEA and DHEAS serum levels have been shown to immediately decrease in multiple trauma patients, indicating an early trauma-related reduction of adrenal androgen synthesis [11,20]. In the further clinical course, a stepwise recovery towards pre-traumatic levels over a period of several months was described. Interestingly, this recovery was significantly influenced by the medication over the clinical course, e.g., with a relevant delay in case of opioid treatment [11]. The decline of DHEAS was both, more pronounced and permanent compared to DHEA, suggesting an additional downregulation of DHEA sulfation after trauma [11,108]. As particularly low levels of DHEAS have been associated with an impaired immune function and higher complications rates (e.g., infections, mortality) [11,109], substitution of DHEAS or DHEA with stimulation of its sulfation might represent promising approaches for a beneficial modulation of the posttraumatic course. This is also true for septic patients who also demonstrated a significant reduction of DHEAS levels [108].

Cortisol and DHEAS appear to be the antagonists. DHEAS has the potential to counteract the immunosuppressive effect of cortisol. In post-traumatic and septic conditions, a decrease in DHEAS has been shown to enhance the immunosuppressive effect of cortisol [110].

Under additional consideration of the positive modulatory influence of DHEA on the immune system without proven significant adverse effects in experimental studies, a clinical utilization of DHEA or DHEA-S within the framework of controlled test conditions, for example, in septic or trauma patients, would be a possible step to prove the efficiency of this new therapeutic approach. However, interventional clinical studies so far only exist for healthy volunteers and patients with autoimmune disorders or osteoporosis who showed beneficial effects on immune response or the underlying disease (e.g., reduced inflammatory response, increased bone mass) [12,13]. For practicality of DHEA supplementation in case of trauma or sepsis, specific considerations in terms of dosing, delivery and safety under specific posttraumatic or -infectious conditions (e.g., polypharmacy, impaired hepatic function) are obligatory needed (Table 1).

## 5. Sympathetic-Adrenergic (SAS) and Parasympathetic-Cholinergic (PCS) System

The immunomodulatory effect of the activation of the sympatho-adrenergic system (SAS) by catecholamines has been known for a long time and has been proven by several animal and human studies. Likewise, the expression of α- as well as β-adrenergic receptors, those in a higher density, on the surface of nearly all cells of the immune system could be validated [111]. The subcutaneous or intravenous administration of the catecholamines adrenalin and noradrenalin leads to marked changes of the migration behavior and activity of circulating T- and NK-cells in healthy volunteers. This effect was inhibited by the administration of a non-selective but not of a β1-selective antagonist, which indicates that the effect might be mediated by β2-receptors [112]. Even after release of endogenous catecholamines, similar effects could be found [111]. Furthermore, it could be shown that catecholamines influence the release of pro- (IL-1β, IL-2, IL-6, IL-12, TNF-α) as well as anti-inflammatory cytokines (IL-10), and these effects could also be inhibited by the administration of β-adrenergic antagonists. Considering a synopsis of current literature, concerning TH1-cytokines, a rather inhibiting impact could be observed whereas with the release of TH2-cytokines, an activating impact seems to predominate [111,113]. Overall, catecholamines in vitro inhibit the adaptive immunity by reducing the proliferation of T helper, T cytotoxic and B-cells and shifting the TH1/TH2 balance towards Th2 cells [114]. In this context, it should be mentioned that the catecholamine dopamine is a potent inhibitor of the MDSC-mediated immunosuppression via the DA and D1-like receptors [115]. MDSCs have been shown to play a central role in the regulation of the pro-inflammation response in the early stage of sepsis. Their function seems to be the limitation of hyperinflammation by L-arginine degradation, production of ROS and NO, the secretion of anti-inflammatory cytokines like IL-10, inducing apoptosis mediated by FAS-FASL, and the activation of T regulatory cells (Tregs) [116,117,118,119]. On the other hand, this anti-inflammatory role seems to be disadvantageous in the later course of sepsis [120]. A function of MDSCs that has been insufficiently investigated so far involves the cell–cell crosstalk with macrophages, the induction of an M2 phenotype and the associated influence of MDSCs on the resolution of inflammation [121]. In the context of a consecutive chronic critical illness and a persistent inflammation immunosuppression and catabolism syndrome, MDSCs appear to be essential for the preservation of existing immunosuppression by suppression of the lymphocyte proliferation [120].

It was shown that adrenaline and noradrenaline are able to influence the release of pro- and anti-inflammatory cytokines, and it was postulated that both catecholamines are involved in the dysregulation of the immune system [111]. Accordingly, an animal model of hemorrhagic shock showed that the shock-induced mobilization of immune cells is blocked independently of the blood pressure by pharmacological blocking of adrenergic β-receptors. Additionally, an increased apoptosis rate of splenocytes was documented [122]. In another study using a murine sepsis model, the pharmacological blocking of adrenergic β-receptors led to an inhibition of the proliferation as well as an increased apoptosis rate of splenocytes and a varied release of cytokines (IFN-γ, IL-6). These immunological effects were accompanied by an increased mortality rate and a deteriorated clinical situation of the mice [123]. Similar results have been shown after selective inhibition of β2-receptors in a mouse sepsis model [122,124]. The exogenous infusion of adrenaline led to significant changes of the distribution of immune cells in the blood of septic mice. The apoptosis and proliferation rate of splenocytes and the release of cytokines were also significantly different without effects on the mortality rate or clinical situation. Additional administration of propranolol intensified the adrenergic effects on the apoptosis of immune cells, antagonized the adrenalin-induced cell mobilization and led to an increased release of IL-6. These effects were accompanied by a markedly increased mortality. Further studies about the pharmacological blocking of adrenergic β-receptors in animal models of sepsis prove a modulation of the release of pro- and anti-inflammatory cytokines even though the results are partially controversial [124].

The impact of circulating catecholamines or the blocking of adrenergic receptors on the immune system could also be shown under conditions of hemorrhagic shock and sepsis. Most of these data are based on animal studies, but in a small prospective study in trauma patients, it could be shown that administration of β-receptor blockers decreased serum IL-6 levels and that patients pretreated with β-receptor blockers had lower initial base deficits after trauma [125]. Clinical studies concerning the immunomodulatory effects of catecholamines in multiple-injured or septic patients are rare, because the use of hypotensive drugs, such as β-blockers in severe sepsis and septic shock, raises justified safety issues. Beyond that, the administration of catecholamines or of β-adrenergic antagonists is often clinically without alternative. A retrospective study reported that critically ill trauma patients receiving β-blockers had a significantly lower in-hospital mortality compared to patients with similar ISS scores not receiving β-blockers (11% vs. 19%) [126]. In general, β-blockers are used in sepsis under the intention to modulate the cardiovascular system but not to influence the inflammatory response; nonetheless β-blockade resulted in a decreased 28 day mortality in septic patients treated with esmolol [127]. Critically ill septic patients with chronic β-blocker prescriptions had lower 28 day mortality than sensitivity and pair-matched controls [128]. These improved outcomes with β-blockers could be due to decreased myocardial oxygen demand [129], improved myocardial oxygen utilization [130] and/or immunomodulation of hypercatecholaminemia [131].

Although several beneficial effects of β-blockers in trauma and sepsis have been described, including restoration of normal cellular metabolism, improved glucose regulation and improved cardiac function [132], the consequences of this interaction for the clinical treatment of patients after multiple trauma or during sepsis in terms of immunomodulation are not clear. The effects of β- blockade on infectious outcomes following the systemic inflammatory response syndrome (SIRS) [133] and the compensatory anti-inflammatory response syndrome (CARS) [134] are unknown. Therefore, more basic research is needed to elucidate the intra- and extracellular mechanisms of immunomodulation of β-blockers. Further, it has to be determined which patients may benefit and especially at which timepoint in the treatment course since an initial sympathetic activation after injury is beneficial but a persistent severe overactivation detrimental. Therefore, the immune suppressive side effects of the β-adrenergic antagonists should be critically included in the therapy decision.

Besides the sympathetic-adrenergic system, also the parasympathetic-cholinergic system (PCS) is able to modulate the inflammatory response [135,136]. The activation of the parasympathetic-cholinergic system via the release of the neurotransmitter acetylcholine (ACh) results in an immune-suppression by inhibition of cytokine production [137]. ACh binds to both nicotinic and muscarinic cholinergic receptors. The main nicotinic cholinergic receptor found on macrophages is the α7 nicotinic ACh receptor subunit (α7nAChR) [137]. It is believed that cholinergic agonists through the activation of α7nACh receptors inhibit NF-κB activation and hence downregulate the production of pro- inflammatory cytokines, such as TNFα [137]. Cholinergic stimulation has been shown to reduce pro- inflammatory cytokine production and prevent lethal tissue injury in multiple models of local and systemic inflammation and sepsis, including acute lung injury, hemorrhagic shock or polymicrobial sepsis [136]. These findings encouraged researchers to assess the therapeutic potential of vagus nerve stimulation (VNS) in attenuating the systemic inflammatory responses evoked by endotoxemia [137]. Direct electrical stimulation of the peripheral vagus nerve in vivo during lethal endotoxemia in rats inhibited TNFα synthesis in the liver, attenuated peak serum TNFα amounts and prevented the development of shock [137]. A beneficial effect of VNS immunomodulation has been reported in other studies for different immunological pathologies [137]. Interestingly, the immunomodulatory effect of vagus nerve stimulation in terms of systemic TNFα reduction is dependent on the spleen, since it fails to work in splenectomized animals [136]. Interruption of the common celiac branch of the abdominal vagus nerve abolishes vagal anti-inflammatory effects, suggesting that cholinergic signaling targets the spleen via this specific branch of the vagus nerve [136].

To date, there are no human studies reporting on the specific pharmacological stimulation of the parasympathetic-cholinergic system in order to modulate the inflammatory response in trauma or septic patients. Nonetheless, transcutaneous mechanical vagus nerve stimulation can exhibit anti-inflammatory effects that may be considered in the clinical setting under special circumstances (Table 1).

## 6. Conclusions

With the exception of corticosteroids, up-to-date proactive modulation of the inflammatory response following trauma and sepsis has not been introduced into clinical practice. Although animal studies show great potential of neuroendocrine immune modulation following trauma and sepsis, knowledge concerning dosage and timing in clinical practice remains unclear. Especially the potential of severe side-effects caused by modulation of this highly complex and dynamic inflammatory response cannot be predicted so far. Future studies are required to achieve the transfer of promising data from bench to bedside.

## Figures and Tables

**Table 1 jcm-09-02287-t001:** Clinical studies.

**HPA Axis**					
**Insult**	**Author**	**Type of Study**	**Number of Included Patients**	**Medication/Substance**	**Result**
Sepsis	Sprung et al.	multicenter, randomized, double-blind, placebo-controlled	499	Hydrocortisone	no improvement of survival
					no improvement of reversal of shock
Sepsis	Lian et al.	meta-analysis	10,044	Corticosteroids	reduction of 28-day mortality
Sepsis	Fang et al.	meta-analysis	9564	Corticosteroids	reduction of 28-day mortality
Trauma	Roquilly et al.	multicenter, randomized, double-blind, placebo-controlled	149	Hydrocortisone	reduction of hospital-acquired pneumonia
**HPG Axis**					
**Insult**	**Author**	**Type of Study**	**Number of Included Patients**	**Medication/Substance**	**Result**
Surgical patients	Bulger et al.	randomized, double-blind, placebo-controlled	41	Oxandrolone	prolonged CMV, no improved outcome
Trauma/Sepsis	Frink et al.	monocenter, comparative study	143	none	lower MODS and sepsis rates in females
Trauma/Sepsis	Offner et al.	monocenter, comparative study	545	none	male gender is associated with infections
Trauma/Sepsis	Trentzsch et al.	retrospective, multicenter, comparative study	29,353	none	male gender is associated with higher MOF, sepsis and mortality rates
Trauma/Sepsis	Trentzsch et al.	retrospective, multicenter, comparative study	10,334	none	female gender is associated with improved organ function and lower sepsis rates
Trauma/Sepsis	Schoeneberg et al.	retrospective, multicenter, comparative study	962	none	female gender is associated with lower sepsis rates
Surcical patients	Wichmann et al.	monocenter, comparative study	40	none	lower IL-6 concentrations in women compared to men after surgery
Surgical patients/Sepsis	Wichmann et al.	monocenter, comparative study	4218	none	lower sepsis incidence in women
Trauma/Sepsis	Deitch et al.	monocenter, comparative study	5192	none	hormonally active women had a better physiological response
Trauma	Haider et al.	retrospective, multicenter, comparative study	48,394	none	hormonally active women showed an improved survival
Trauma	Wohltmann	retrospective, multicenter, comparative study	20,261	none	hormonally active womed showed an improved survival
Trauma	Oberholzer et al.	monocenter, comparative study	1276	none	females showed an decreased posttraumatic morbidity
Trauma	George et al.	retrospective, multicenter, comparative study	155,691	none	females showed an improved survival
Trauma/Sepsis	Rappold et al.	monocenter, comparative study	1229	none	female gender offered no protection from ARDS, pneumonia, sepsis or death
Trauma	Gannon et al.	retrospective, multicenter, comparative study	22,332	none	no influence of female gender on outcome
Sepsis	Eachempati et al.	prospective, monocenter, comparative study	1348	none	female gender is an independent predictor of increased mortality
Trauma	Napolitano et al.	monocenter, comparative study	18,892	none	no influence of female gender on outcome
Trauma	Dossett et al.	prospective, multicenter, comparative study	991	none	serum estradiol levels were a marker for injury severity and a predictor of death
Sepsis	Sakr et al.	prospective, multicenter, comparative study	3902	none	women had a lower sepsis prevalence
**DHEA**					
**Insult**	**Author**	**Type of Study**	**Number of Included Patients**	**Medication/Substance**	**Result**
Influence of trauma on systemic DHEA/DHEAS levels	Foster et al.	prospective, monocenter, comparative study	95 patients	DHEA/DHEAS	Rapid decrease of DHEA after trauma with recovery to the normal range by 3 months DHEAS rapidly reduced after trauma with no recovery until 6 months after trauma
Influence of trauma on systemic DHEA/DHEAS levels	Brorsson et al.	prospective, monocenter, comparative study	50 patients	DHEA/DHEAS	Significant decrease of DHEA and DHEAS over the observation period of 5 days
Effects of sepsis and trauma on DHEA/DHEAS after ACTH stimulation	Arlt et al.	prospective, monocenter, comparative study	212 patients	DHEA/DHEAS	Decreased DHEAS levels after trauma and sepsis Increased DHEA levels after sepsis, decreased levels after trauma No increase of DHEA after ATCH-application in septic patients
**SAS/PCS**					
**Insult**	**Author**	**Type of Study**	**Number of Included Patients**	**Medication/Substance**	**Result**
Sepsis	Macchia et al.	retrospective, multicenter, comparative study	9465	beta-blocker	reduction of 28-day mortality
Sepsis	Morelli et al.	prospective, randomized, monocenter, comparative study	154	beta-blocker (esmolol)	reduction of mortality
Trauma	Bukur et al.	monocenter, retrospective, comparative study	663	beta-blocker	reduction of mortality

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
