# Peer review of "Neuroendocrine Modulation of the Immune Response after Trauma and Sepsis: Does It Influence Outcome?"

_jcm, 2020, doi:10.3390/jcm9072287_

Round 1

Reviewer 1 Report

The review article submitted by Kobbe and colleagues discusses our current understanding of how activation of the neuroendocrine system in the settings of trauma and sepsis may influence patient outcome via its effects upon the immune system. Reviewing data from both rodent and human-based studies, the authors discuss how infection and post-injury induced changes in the circulating levels of glucocorticoids, sex steroids, the adrenal androgen DHEA (and its downstream metabolites) as well as adrenaline/nor-adrenaline may modulate the immune response.

My comments on the manuscript are as follows:

Introduction

  1. There are several sentences in the introduction that include statements that need to be supported by a citation to published work. These sentences are:

(a)"With regard to the HPG-axis, which is likewise controlled by the release of hormones of the hypothalamus, decreased testosterone levels could be found in men after severe trauma and during sepsis whereas women react with an increase of their estrogen levels"

(b) "Blood levels of the steroid hormone Dehydroepiandrosterone (DHEA) and its sulphated pattern (DHEA-S) are significantly decreased in critically ill and septic patients"

    2. The introduction should conclude with the aim/purpose of the review            article.

HPA axis

  1. Would the authors consider including a discussion in this section on whether the systemic inflammatory response syndrome that occurs following trauma/sepsis could promote immune suppression (e.g. reduced inflammatory cytokine production) by raising intracellular cortisol concentrations within immune cells that would occur as a consequence of an inflammation-induced increase in the expression/activity of 11B-HSD1? I am thinking in particular about monocytes, in which 11B-HSD1 expression has been shown to be induced following in vitro culture with inflammatory agonists and also in macrophages, which express high levels of this enzyme.
  2. This section contains the sentence..."additionally, glucocorticoids inhibit the function of neutrophil and eosinophil granulocytes as well as macrophages [15]." Reference 15 does not contain any information on the effect of glucocorticoids on macrophage function. The authors should include a specific macrophage reference alongside reference 15, which is granulocyte specific.

HPG axis

1. There appears to be a grammatical error in the following sentence from this section (do the authors mean humoral and not humeral?) -"Furthermore, sex hormones significantly modulated cellular and humeral immune functions after TH?"

2. There are some sentences in this section that include statements that need to be supported by a citation to published work. These sentences are:

  1. "On the one hand, traumatic and infectious insults induce a central suppression of the HPG axis with reduced gonadal androgen production and an associated state of catabolism (reduction of muscle mass, increased nitrogen loss)."
  2. "For example, E2 was shown to be beneficial for cardiac function after trauma-hemorrhage (TH) (e.g. increased cardiac output) due to protective effects on myocardial mitochondria, induction of heme oxygenase-1 (HO-1), downregulation of nuclear factor kappa-light-chain-enhancer of activated B-cells (NF-κB) and the modulation of inflammatory mediators, such as heat shock proteins (HSP) and cytokines."
  3. "The expression and activity of these enzymes were modulated by TH and differed between males and proestrus females. In males, TH resulted in increased levels of 5a-reductase which catalyses the synthesis of 5a-dihydrotestosterone (5a-DHT). 5a-DHT itself is well known to exert even more pronounced immunosuppressive effects than testosterone. In contrast, in proestrus females an enhanced aromatase activity was observed which resulted in an increased E2 synthesis."
  4. "After infections, human mononuclear cells from males were shown to produce lower levels of type I interferons (IFN) in response to TLR-7 ligands and higher IL-10 in response to TLR-9 ligands as compared to females."

DHEA

1. This section contains the sentence..."Furthermore, a deficit of DHEA/DHEAS was associated with an impaired natural killer cell cytotoxic function [71]. The authors should cite the original manuscript that reported this observation and not a review article.

2. In this section, the authors mention how in vitro DHEAS treatment has been shown to increase ROS production by neutrophils. In this cited paper, it was also shown that neutrophils are the only immune cell in the circulation that possess the transporter for DHEAS (OATP-D), suggesting that the ability to respond to circulating DHEAS in vivo is unique to neutrophils. The authors should mention this specificity and also how given that neutrophils do not express steroid sulfatase, the effects of DHEAS will be direct and not a consequence of its conversion to downstream metabolites.   

3. Published work has demonstrated that for neutrophil function, incubation with DHEAS can counteract the immune suppressive effects of cortisol. Given that trauma and sepsis are associated with increased cortisol levels and reduced DHEA(S) levels, the authors should emphasise that the immune suppressive effects of cortisol in these patients would be exacerbated by the concurrent decline in DHEA(S).  

Additional considerations

  1. Studies have demonstrated that glucocorticoids and catecholamines can drive the expansion of myeloid derived suppressor cells (MDSCs). Known for their immune suppressive actions (particularly in the context of sepsis), I feel that this review article would benefit from the inclusion of a discussion on the concept that a neuro-endocrine driven expansion in MDSCs could be a potential mechanism underlying the immune suppression that occurs as a consequence of neuro-endocrine activation in septic/trauma patients.
  2. In the article, the authors did not discuss the parasympathetic nervous system (PNS) in the context of sepsis or trauma-induced immune suppression. A variety of immune cells are known to express the receptor for acetylcholine and exposure to this neurotransmitter has been shown to alter the function of innate and adaptive immune cells. As an anti-inflammatory pathway, the persistent activation of the PNS that may occur in trauma/septic patients (as a consequence of a systemic elevation in circulating pro-inflammatory cytokines) is a potential mechanism that may drive peripheral immune suppression and increase the risk of infection in these patient groups. The authors should consider including a summary of our current understanding of immune regulation by the PNS in the context of sepsis/trauma.
  3. The article would benefit from a figure or table that summarises the known immune modulatory effects of the different neuro-endocrine modulators discussed in the review.

Author Response

Many thanks for the review and the opportunity to edit and improve our manuscript. Enclosed you will find our point-to-point answers to your suggestions. Additionally we have marked the changes in our manuscript in colour.

We would be pleased about an acceptance of the manuscript and remain with kind regards,

The authors

Introduction:

  • There are several sentences in the introduction that include statements that need to be supported by a citation to published work. These sentences are:

  • "With regard to the HPG-axis, which is likewise controlled by the release of hormones of the hypothalamus, decreased testosterone levels could be found in men after severe trauma and during sepsis whereas women react with an increase of their estrogen levels"We have added the reference. The reference was highlighted in color.

  • "Blood levels of the steroid hormone Dehydroepiandrosterone (DHEA) and its sulphated pattern (DHEA-S) are significantly decreased in critically ill and septic patients"We have added the reference. The reference was highlighted in color. 
  • The introduction should conclude with the aim/purpose of the review article
  1. We have added the following paragraph:

“The aim of this review is on the one hand to highlight current insights on how neuroendocrine released messengers are responsible for immunomodulation following severe trauma and during sepsis and on the other hand whether this knowledge has been transferred into clinical practice.“

HPA axis:

  • Would the authors consider including a discussion in this section on whether the systemic inflammatory response syndrome that occurs following trauma/sepsis could promote immune suppression (e.g. reduced inflammatory cytokine production) by raising intracellular cortisol concentrations within immune cells that would occur as a consequence of an inflammation-induced increase in the expression/activity of 11B-HSD1? I am thinking in particular about monocytes, in which 11B-HSD1 expression has been shown to be induced following in vitro culture with inflammatory agonists and also in macrophages, which express high levels of this enzyme.We added the following paragraph:
  1.  

“Recently, the role of 11b-Hydroxysteroid dehydrogenase type 1 (11b-HSD1) in acute and chronic inflammation has been pointed out [21-23]. 11b-HSD1 causes an intracellular conversion of inactive cortisone to the active cortisol. Therefore, 11b-HSD1 is an intracellular gate-keeper for glucocorticoid action [22]. Interestingly, the expression of 11b-HSD1 is greatly up-regulated during differentiation of monocytes into macrophages thus theoretically curbing the inflammatory potency of these cells [21]. However, it appears that this intracellular immunomodulation by 11b-HSD1 is disturbed during trauma and hemorrhage resulting in an inefficacy of released glucocorticoids to modulate the inflammatory response [24].”

  • This section contains the sentence..."additionally, glucocorticoids inhibit the function of neutrophil and eosinophil granulocytes as well as macrophages [15]." Reference 15 does not contain any information on the effect of glucocorticoids on macrophage function. The authors should include a specific macrophage reference alongside reference 15, which is granulocyte specific.

We have added the reference. The reference was highlighted in color.

HPG axis:

  • There appears to be a grammatical error in the following sentence from this section (do the authors mean humoral and not humeral?) -"Furthermore, sex hormones significantly modulated cellular and humeral immune functions after TH?"We have meant humoral and corrected it.
  • There are some sentences in this section that include statements that need to be supported by a citation to published work. These sentences are:
  • "On the one hand, traumatic and infectious insults induce a central suppression of the HPG axis with reduced gonadal androgen production and an associated state of catabolism (reduction of muscle mass, increased nitrogen loss)."

We have added the reference. The reference was highlighted in color.

  • "For example, E2 was shown to be beneficial for cardiac function after trauma-hemorrhage (TH) (e.g. increased cardiac output) due to protective effects on myocardial mitochondria, induction of heme oxygenase-1 (HO-1), downregulation of nuclear factor kappa-light-chain-enhancer of activated B-cells (NF-κB) and the modulation of inflammatory mediators, such as heat shock proteins (HSP) and cytokines."

We have added the reference. The reference was highlighted in color.

  • "The expression and activity of these enzymes were modulated by TH and differed between males and proestrus females. In males, TH resulted in increased levels of 5a-reductase which catalyses the synthesis of 5a-dihydrotestosterone (5a-DHT). 5a-DHT itself is well known to exert even more pronounced immunosuppressive effects than testosterone. In contrast, in proestrus females an enhanced aromatase activity was observed which resulted in an increased E2 synthesis."

We have added the reference. The reference was highlighted in color.

  • "After infections, human mononuclear cells from males were shown to produce lower levels of type I interferons (IFN) in response to TLR-7 ligands and higher IL-10 in response to TLR-9 ligands as compared to females."

We have added the reference. The reference was highlighted in color.

DHEA:

  • This section contains the sentence..."Furthermore, a deficit of DHEA/DHEAS was associated with an impaired natural killer cell cytotoxic function [71]. The authors should cite the original manuscript that reported this observation and not a review article.We have added the reference. The reference was highlighted in color.
  • In this section, the authors mention how in vitro DHEAS treatment has been shown to increase ROS production by neutrophils. In this cited paper, it was also shown that neutrophils are the only immune cell in the circulation that possess the transporter for DHEAS (OATP-D), suggesting that the ability to respond to circulating DHEAS in vivo is unique to neutrophils. The authors should mention this specificity and also how given that neutrophils do not express steroid sulfatase, the effects of DHEAS will be direct and not a consequence of its conversion to downstream metabolites.   We added the following paragraph:

Neutrophils hold a unique position among the leukocytes, since they are the only subpopulation with an active transporter, the organic anion-transporting polypeptide D (OATP-D). Furthermore, neutrophils do not have steroid sulfatase, which activates DHEAS to DHEA. The effect of DHEA consequently must be a direct one.

  • Published work has demonstrated that for neutrophil function, incubation with DHEAS can counteract the immune suppressive effects of cortisol. Given that trauma and sepsis are associated with increased cortisol levels and reduced DHEA(S) levels, the authors should emphasise that the immune suppressive effects of cortisol in these patients would be exacerbated by the concurrent decline in DHEA(S).We added the following paragraph:  

“Cortisol and DHEAS appear to be the antagonists. DHEAS has the potential to counteract the immunosuppressive effect of cortisol. In post-traumatic and septic conditions, a decrease in DHEAS has been shown to enhance the immunosuppressive effect of cortisol [110].”

Additional considerations:

  • Studies have demonstrated that glucocorticoids and catecholamines can drive the expansion of myeloid derived suppressor cells (MDSCs). Known for their immune suppressive actions (particularly in the context of sepsis), I feel that this review article would benefit from the inclusion of a discussion on the concept that a neuro-endocrine driven expansion in MDSCs could be a potential mechanism underlying the immune suppression that occurs as a consequence of neuro-endocrine activation in septic/trauma patients.We added the following paragraphs:  

“In the search for the mechanisms of the immunomodulating effect of estrogen, myeloid-derived suppressor cells (MDSC), a heterogeneous population of the myeloid lineage, which modulate the adaptive immune response, have attracted the focus of research efforts [76]. These cell subtypes have been shown to constitute a crucial component of the innate immune system in various inflammatory states. During systemic inflammation, MDSCs are recruited from the bone marrow [77]. In this respect, it could be shown in different mouse models that after CLP procedure and recruitment, an accumulation of MDSCs takes place in secondary lymphatic organs. The recruitment of MDSCs is mediated by MAMPs (e.g. LPS) and DAMPs (e.g. HMGB1) [77-79]. Furthermore, MDSC are induced and activated in the presence of estrogen and cytokines, such as IL-6, IFN-γ, and IL-1β, and strongly contribute to T-cell dysfunction in various diseases such as sepsis, tumorgenesis and trauma [80-82]. Especially in the case of estrogen, a direct activation of the STAT3 signaling pathway and upregulation of JAK2 and SRC in MDSCs and a consecutive anti-inflammatory function of MDSCs could be demonstrated [83].”

“In this context, it could be shown that the GR receptor is also expressed on circulating MDSCs. Lu et al. could show in an experimental study on a liver injury mouse model that the modulating effect of glucocorticoids is caused by suppression/activation of HIF1α and HIF1α-dependent glycolysis [104]. Based on these observations, the modulation of MDSC function by systemic steroids may represent a new therapeutic target, although detailed data on the timing of use and type of steroid are still lacking. Furthermore, the function of MDSC in sepsis and trauma has not been sufficiently studied [105, 106]. The vast majority of data are based on studies in tumors, although an increasing number of studies are highlighting the role of MDSC subtypes in the resolution of inflammation after severe sepsis and trauma [106, 107].”

“In this context it should be mentioned that the catecholamine dopamine is a potent inhibitor of the MDSC-mediated immunosuppression via the DA and D1-like receptors [115]. MDSCs have been shown to play a central role in the regulation of the pro-inflammation response in the early stage of sepsis. Their function seems to be the limitation of hyperinflammation by L-arginine degradation, production of ROS and NO, the secretion of anti-inflammatory cytokines like IL-10, inducing apoptosis mediated by FAS-FASL, and the activation of T regulatory cells (Tregs) [116-119]. On the other hand, this anti-inflammatory role seems to be disadvantageous in the later course of sepsis [120]. A function of MDSCs that has been insufficiently investigated so far involves the cell-cell crosstalk with macrophages, the induction of an M2 phenotype and the associated influence of MDSCs on the resolution of inflammation [121]. In the context of a consecutive chronic critical illness and a persistent inflammation immunosuppression and catabolism syndrome, MDSC appear to be essential for the preservation of existing immunosuppression by suppression of the lymphocyte proliferation [120].”

Reviewer 2 Report

The authors have presented a review to highlight the current knowledge of the neuroendocrine modulation of the immune system during trauma and sepsis. 

I have the following points for consideration by the authors.

  1. Post traumatic changes in the immune system are crucial to the development of complications in multiply injury patients. Please comment on the duration and degree of activation of the SAS system following trauma. It is not only the activation which is part of the fight or flight response but rather the duration of activation and degree of elevation of the SAS that impacts outcomes. 
  2. In addition to trauma and sepsis being impacted by neuroendocrine modulation please consider this impact on Burn patients as well. 
  3. Also, please review Loftus et al. Shock 2016 46:341-351 for the role of beta blockade after trauma and sepsis and the benefits to immunomodulation. 

Author Response

Many thanks for the review and the opportunity to edit and improve our manuscript. Enclosed you will find our point-to-point answers to your suggestions. Additionally we have marked the changes in our manuscript in colour.

We would be pleased about an acceptance of the manuscript and remain with kind regards,

the authors

All changes are marked in colour!

We added the paragraph:

"A retrospective study reported that critically ill trauma patients receiving β-blockers had a significantly lower in-hospital mortality compared to patients with similar ISS scores not receiving β-blockers (11% vs. 19%) [126]. In general, β-blockers are used in sepsis under the intention to modulate the cardiovascular system but not to influence the inflammatory response; nonetheless β-blockade resulted in a decreased 28 day mortality in septic patients treated with esmolol [127]. Critically ill septic patients with chronic β-blocker prescriptions had lower 28 day mortality than sensitivity and pair-matched controls [128]. These improved outcomes with β-blockers could be due to decreased myocardial oxygen demand [129], improved myocardial oxygen utilization [130], and/or immunomodulation of hypercatecholaminemia [131].

Although several beneficial effects of β-blockers in trauma and sepsis have been described, including restoration of normal cellular metabolism, improved glucose regulation, and improved cardiac function [132], the consequences of this interaction for the clinical treatment of patients after multiple trauma or during sepsis in terms of immunomodulation are not clear. The effects of β- blockade on infectious outcomes following the systemic inflammatory response syndrome (SIRS) [133] and the compensatory anti-inflammatory response syndrome (CARS) [134] are unknown. Therefore, more basic research is needed to elucidate the intra- and extracellular mechanisms of immunomodulation of β-blockers. Further, it has to be determined which patients may benefit and especially at which timepoint in the treatment course since an initial sympathetic activation after injury is beneficial but a persistent severe overactivation detrimental. Therefore, the immune suppressive side effects of the β-adrenergic antagonists should be critically included in the therapy decision.

Besides the sympathetic-adrenergic system also the parasympathetic-cholinergic system (PCS) is able to modulate the inflammatory response [135, 136]. The activation of the parasympathetic-cholinergic system via the release of the neurotransmitter acetylcholine (ACh) results in an immune-suppression by inhibition of cytokine production [137]. ACh binds to both nicotinic and muscarinic cholinergic receptors. The main nicotinic cholinergic receptor found on macrophages is the α7 nicotinic ACh receptor subunit (α7nAChR) [137]. It is believed that cholinergic agonists through the activation of α7nACh receptors inhibit NF-κB activation and hence downregulate the production of pro- inflammatory cytokines, such as TNFα [137]. Cholinergic stimulation has been shown to reduce pro- inflammatory cytokine production and prevent lethal tissue injury in multiple models of local and systemic inflammation and sepsis, including acute lung injury, hemorrhagic shock, or polymicrobial sepsis [136]. These findings encouraged researchers to assess the therapeutic potential of vagus nerve stimulation (VNS) in attenuating the systemic inflammatory responses evoked by endotoxemia [137]. Direct electrical stimulation of the peripheral vagus nerve in vivo during lethal endotoxaemia in rats inhibited TNFα synthesis in the liver, attenuated peak serum TNFα amounts, and prevented the development of shock [137]. A beneficial effect of VNS immunomodulation has been reported in other studies for different immunological pathologies[137]. Interestingly, the immunomodulatory effect of vagus nerve stimulation in terms of systemic TNFα reduction is dependent on the spleen, since it fails to work in splenectomized animals [136]. Interruption of the common celiac branch of the abdominal vagus nerve abolishes vagal anti-inflammatory effects, suggesting that cholinergic signaling targets the spleen via this specific branch of the vagus nerve [136].

Up-to-date, there are no human studies reporting on the specific pharmacological stimulation of the parasympathetic-cholinergic system in order to modulate the inflammatory response in trauma or septic patients. Nonetheless, transcutaneous mechanical vagus nerve stimulation can exhibit anti-inflammatory effects that may be considered in the clinical setting under special cirumstances."